# Gene Expression Changes Induced by Exposure of RAW 264.7 Macrophages to Particulate Matter of Air Pollution: The Role of Endotoxins

**DOI:** 10.3390/biom12081100

**Published:** 2022-08-10

**Authors:** Adam Roman, Michał Korostyński, Monika Jankowska-Kieltyka, Marcin Piechota, Jacek Hajto, Irena Nalepa

**Affiliations:** 1Department of Brain Biochemistry, Maj Institute of Pharmacology, Polish Academy of Sciences, 31-343 Kraków, Poland; 2Department of Molecular Neuropharmacology, Maj Institute of Pharmacology, Polish Academy of Sciences, 31-343 Kraków, Poland

**Keywords:** particulate matter, macrophages, metabolic activity, microarray analysis, gene expression profiling

## Abstract

Despite the variable chemical and physical characteristics of particulate air pollutants, inflammation and oxidative stress have been identified as common mechanisms for cell damage and negative health influences. These effects are produced by organic components, especially by endotoxins. This study analyzed the gene expression profile after exposure of RAW 264.7 cells to the standard particulate matter (PM) material, NIST1648a, and PM with a reduced organic matter content, LAp120, in comparison to the effects of lipopolysaccharide (LPS). The selected parameters of cell viability, cell cycle progression, and metabolic and inflammatory activity were also investigated. Both forms of PM negatively influenced the parameters of cell activity. These results were generally reflected in the gene expression profile. Only NIST1648a, excluding LAp120, contained endotoxins and showed small but statistically significant pro-inflammatory activity. However, the gene expression profiling revealed strong pro-inflammatory cell activation induced by NIST1648a that was close to the effects of LPS. Changes in gene expression triggered by LAp120 were relatively small. The observed differences in the effects of NIST1648a and LAp120 were related to the content of organic matter in which bacterial endotoxins play an important role. However, other organic compounds and their interactions with other PM components also appear to be of significant importance.

## 1. Introduction

Particulate matter (PM) of air pollution is currently considered to be one of the most important threats to human health and life. Studies conducted over the last several decades have shown that exposure to PM shortens life expectancy, increases mortality and adversely affects the course of many diseases, mainly those associated with the respiratory and cardiovascular systems [1,2,3]. With the growth in the human population, industrial development and urbanization, air pollution with PM is also increasing. This problem affects a large part of the human population, as elevated PM levels tend to be found in densely populated areas. The health effects of exposure and the mechanisms of the negative impacts of PM on the body are not yet fully understood due to the extremely high variability in the chemical composition of pollutants, their physical properties, changes in the environment and the interaction of all these factors in influencing metabolic processes [4,5,6].

PM is a combination of solid particles and liquid droplets of various origins, sizes and chemical compositions [5,6]. PM consists of particles of natural (e.g., soil mineral components, sea salt, plant pollen and fungal spores, microorganisms and their fragments) and anthropogenic origins (e.g., soot, ash and particles released as a result of technological processes, transport and fuel combustion). PM demonstrates exceptional complexity and variability in its chemical composition and physical properties, depending on the emission source, season and weather conditions. In urbanized areas, components from an anthropogenic origin constitute the majority of the total PM mass [6,7]. Both natural and anthropogenic PMs are composed of organic and inorganic components. Both of these categories of ingredients have deleterious effects; however, the mechanisms of action and biological effects may be different. In vivo and in vitro clinical studies indicate that the negative effects of the inorganic components are related to the presence of metals, while among the organic components, polycyclic aromatic hydrocarbons (PAHs) and bacterial endotoxins have the greatest impact on biological processes (e.g., [8,9,10,11]).

Endotoxins are lipopolysaccharides (LPS) that are a part of the cell wall of Gram-negative bacteria including enterobacteria. The LPS molecule consists of three main components: lipid A, core polysaccharides and the O antigen. Differences in the chemical composition of individual components determine the diversity of the biological properties of LPS. LPS induces a strong reaction from the immune system, mainly through the activation of Toll-like receptor-4 (TLR4) present on cells of innate immunity: monocytes, macrophages and dendritic cells [12]. The innate immune response that has been triggered in this way involves many signaling molecules including peripherally secreted cytokines and inflammatory mediators. An initiated signaling cascade enhances the immune response and activates specific mechanisms. The immune signaling is also transferred to the brain through the bloodstream. In special circumstances, some cytokines are able to cross the blood–brain barrier (BBB), activating the brain’s stress response system: the hypothalamus–pituitary–adrenal (HPA) axis. Over a dozen years of research has provided increasing evidence showing the involvement of the gut microbiota in brain disease pathogenesis, and LPS is considered to be an important triggering factor involved. For instance, an increase in the translocation of LPS from Gram-negative enterobacteria (leaky gut) plays a role in the inflammatory pathophysiology of depression [13]. On the other hand, it should be emphasized that LPS is a common component of PM, as it is released into the environment as a result of the breakdown of ubiquitous bacterial cells. The LPS content in PM is variable and depends on the location and type of emission sources, weather and season [11,14,15,16,17].

PM penetrates into the body mainly through the respiratory system, where it causes, among others, oxidative stress and pro-inflammatory activation of pulmonary macrophages [18,19]. Macrophages are cells of innate immunity commonly present in all tissues and organs of the body [20]. They are the first line of defense against a wide range of biological and abiotic factors disturbing tissue homeostasis including PM [20,21]. As a result of activation, they ingest (i.e., phagocytose) pathogens, foreign particles and dead or damaged cells, and after removal of the threat, they participate in the processes of tissue regeneration and restoration of their normal functions [20]. As macrophages have a potent influence on the direction and the course of immune reactions [22], their pro-inflammatory activation may affect not only local but also the entire immune system.

Our previous studies showed that the standard urban particulate air pollutant material, NIST1648a, depending on the concentration and duration of exposure, induced the synthesis of reactive oxygen species (ROS) and the release of nitric oxide (NO) as well as increased macrophagal cell death in in vitro cultures [23]. NIST1648a devoid of the majority of its organic compounds, as a result of 2 h of cold plasma treatment, referred to as LAp120, had a greater effect on ROS synthesis and cell death but did not induce NO release. Pro-inflammatory activation of macrophage cells by NIST1648a, manifested by an increase in NO release [24], may be the result of both oxidative stress [25,26] as well as the response to LPS [27].

In vitro studies offer simple and easy models for the assessment of the negative effects of PM on the biological activity of cells, widely used despite several limitations [28]. The macrophage-like cell line, RAW 264.7, is one of the most commonly used cell types in these studies, especially in the context of the pro-inflammatory effects of PM (e.g., [11,16,29,30,31]). In the current study, we investigated whether exposure of RAW 264.7 cells to NIST1648a or LAp120 affected the gene expression profile, and the impact of PM was compared to the effects of LPS. Since biological effects are different depending on the concentration and time of exposure [31], two concentrations of both PM forms (1 and 100 µg/mL) were used and selected parameters of metabolic activity and changes in gene expression were studied after short-term (4–6 h) and long-term (48 h) exposure.

## 2. Materials and Methods

### 2.1. Reagents

Dulbecco’s modified Eagle’s medium (DMEM), Dulbecco’s phosphate-buffered saline with and without calcium and magnesium (PBS), and fetal bovine serum (FBS) were obtained from Gibco (Invitrogen, Poisley, UK). Lipopolysaccharides (LPS) from *E. coli* serotype O111:B4, penicillin–streptomycin solution, propidium iodide (PI), 4,5-diaminofluorescein diacetate (DAF-2 DA), dimethyl sulfoxide (DMSO), RNAse DNAse-free from bovine pancreas, resazurin, sulfanilamide and N-(1-naphthyl) ethylenediamine dihydrochloride were obtained from Sigma-Aldrich (St. Louis, MO, USA). Orthophosphoric acid was purchased from POCh (Gliwice, Poland). NIST1648a was obtained from the National Institute of Standards and Technology (Gaithersburg, MD, USA). The Pierce LAL Chromogenic Endotoxin Quantitation Kit was purchased from Thermo Fisher Scientific (Waltham, MA, USA). Polymyxin B (Poly B) was obtained from InvivoGen (Toulouse, France). Water LAL reagent was obtained from Lonza (Basel, Switzerland). Culture dishes were purchased from Corning (New York, NY, USA) and Nunc (Roskilde, Denmark).

### 2.2. Preparation of the PM Suspensions

NIST1648a is a standard urban air pollutant material with a fully known particle size distribution and chemical composition, which was described in the certificate of analysis [32]. NIST1648a was subjected to 2 h of treatment with cold plasma using the Plasma Zepto system (Diener Electronic GmbH, Ebhausen, Germany), which removes most of the organic matter [33]. Organic components were removed at the Faculty of Chemistry of the Jagiellonian University in Kraków. The acronym, LAp120, was formed for the obtained material with a reduced content of organic carbon (from 9.12 to 1.68 wt%) [23].

Both NIST1648a and LAp120 were suspended in PBS, and the suspensions were sonicated for 3 min in an ultrasonic water bath immediately before being added to cell cultures. Our previous experiments showed that PM at a final concentration of 1 µg/mL of the culture medium caused only slight changes in the viability and metabolic activity parameters of the RAW 264.7 cells after both 4–6 and 48 h of exposure. On the other hand, PM at a final concentration of 100 µg/mL caused clear changes after both short-term (4–6 h) and long-term (48 h) exposure [23]. Thus, the abovementioned concentrations of PM and exposure times were used in this work.

### 2.3. Cell Culture and Treatment

The RAW 264.7 mouse macrophage cell line (obtained from the Department of Immunology, Jagiellonian University Medical College, Kraków, Poland) was cultured under standard conditions (37 °C, 90% humidity, 5% CO_2_) in a culture medium based on DMEM and supplemented with 10% heat-inactivated FBS, 50 U/mL penicillin and 50 µg/mL streptomycin.

The cell suspension, at a concentration of 2.5 × 10^4^ cells per mL, was plated into 6-well culture plates at a volume of 3.2 mL (for gene expression profiling) or 96-well flat-bottomed culture plates at a volume of 0.1 mL (for other assays), and the cultures were incubated for 24 h to stabilize the culture conditions. Suspensions of PM (as well as the LPS solution in the case of gene expression profiling) were added 48 or 4 h before the end of the culture. Control wells were supplemented with PBS. The final concentrations of NIST1648a and LAp120 amounted to 1 and 100 µg/mL, as mentioned above, which corresponded to 0.33 and 33.33 µg/cm^2^ of the bottom surface of the wells, respectively. LPS was added to the cultures destined for gene expression profiling only as a positive control to reach a final concentration of 10 ng/mL of the culture medium. The assessment of the selected parameters was performed after both short-term (4–6 h) and long-term (48 h) exposure. Differentiation of short-term time points in the range of 4–6 h, to our knowledge, does not affect the results of the study, as both our unpublished results and data from the literature [23,31] showed that no significant qualitative changes occurred in in vitro cultures exposed to PM during this time interval.

### 2.4. Viability Assay

Cell viability was assessed cytometrically following supravital staining of cells with PI as previously described [23]. PI is a water-soluble fluorescent probe that diffuses into cells when the plasma membrane has been compromised and stains only damaged cells. This method makes it possible to distinguish live cells from late apoptotic and necrotic cells [34]. Briefly, after a designated exposure time (i.e., 6 or 48 h), cells were vigorously rinsed in ice-cold PBS without Ca^++^ and Mg^++^ ions and transferred to cytometric tubes. Then, the PI solution was added at a concentration of 50 µg/mL, and after 10 min of incubation in the dark, they were analyzed in a FACSCanto II flow cytometer using FACSDiva 8.0.2 software (both BD Biosciences, San Jose, CA, USA). PI fluorescence was induced by blue laser light (488 nm) and measured in the FL-3 channel (670–735 nm). Unstained cells were considered viable, dimly stained as late apoptotic and brightly stained as necrotic.

### 2.5. Metabolic Activity Assay

The metabolic activity related to the respiratory processes of the cells was assessed using the resazurin reduction test, according to a method described elsewhere [35]. Resazurin is mainly metabolized in mitochondria [36], and its metabolic product, fluorescent resorufin, is released into the culture medium [37]. Briefly, after 4 and 48 h of incubation with both forms of PM, 10 μL of resazurin solution in PBS at a concentration of 0.44 mM was added to the cultures. After 1.5 h of incubation in an incubator under standard conditions, the fluorescence of the cultures was measured using a Synergy MX microplate reader working under the control of Gen5 1.11.4 software (both BioTek Instruments, Winooski, VT, USA). The excitation and emission wavelengths were 560 and 590 nm, respectively. The assessment was performed at least in sextuples, and the data are presented as the percentage of the values noted in the control cultures.

### 2.6. Assay for NO Release

The release of NO was determined as the concentration of nitrites in the culture medium using the Griess reaction, as previously described [35]. This assessment was performed only after 48 h of exposure to NIST1648a or LAp120, as inducible nitric oxide synthase (iNOS) is not constitutively present in cells, and the concentration of nitrite in the medium increases after 24 h [38]. Briefly, at the end of exposure period, 50 μL of supernatant was mixed with 30 μL of 1% sulfanilamide dissolved in 5% orthophosphoric acid and 30 μL of 0.1% water solution of N-(1-naphthyl) ethylenediamine dihydrochloride. After a short period of incubation at room temperature, the absorbance was measured at 540 nm using a Synergy MX microplate reader. The dependence of the LPS concentration on NO release by RAW 264.7 cells in the 48 h cultures was investigated in a separate experiment. The assessment was performed in 5–6 replicates, and the data are presented as the percentage of the control values.

The synthesis of NO was also assessed with the direct method using a DAF-2 DA fluorescent probe and a microplate reader [39,40]. Membrane-permeable DAF-2 DA, when added to the culture, penetrates inside the cells, where it is cleaved by intracellular esterases into nonfluorescent DAF-2. This compound specifically binds NO to form highly fluorescent triazole forms. Briefly, RAW 264.7 cultures were exposed for 72 h to NIST1648a or LAp120 at a concentration of 100 µg/mL. After this time, the culture medium was replaced with 100 μL of PBS after a single washing, and 10 μL of 10 µM DAF-2 DA was added to each well. Control cultures were supplemented with a solvent (1% DMSO in PBS). Then, the cell cultures were incubated under standard conditions for 90 min, and the fluorescence (excitation and emission wavelengths: 495 and 515 nm, respectively) was measured using a Synergy MX microplate reader. The assessment was performed in sextuples, and the data are expressed as relative fluorescence units (RFUs).

### 2.7. Assessment of Cell Proliferation and the Cell Cycle

The cell cycle was analyzed using a flow cytometer, according to the method described by Nicoletti et al. [41]. In this method, cells are fixed with ethanol and stained with PI in the presence of RNAse, and the DNA content is measured with a flow cytometer. The DNA content reflects the status of the cell cycle. Briefly, at the time points specified above (i.e., 6 and 48 h), cells were rinsed with cold PBS and dropwise fixed with 100% ethanol on ice. Fixed cells were stored at 4 °C for at least 24 h. Cells were then washed twice with PBS and stained with PI at a final concentration of 150 µg/mL supplemented with RNAse (250 µg/mL) and then incubated in the dark for 30 min. Cell fluorescence was measured in a FACSCanto II cytometer. PI fluorescence was induced by blue laser light (488 nm) and measured in the FL-3 channel (670–735 nm). Elimination of doublets was performed on the basis of the pulse-area-to-pulse-width ratio of the PI fluorescence [42]. Data were analyzed, using FACSDiva 8.0.2 software, as the percentage of events in the gates describing cells in the G0/G1, S and G2/M phases, also including the sub-G1 subpopulation, i.e., apoptotic events. This method can detect an earlier stage of apoptosis than supravital PI staining [34], when DNA defragmentation and condensation occur, but the cell membrane is not yet damaged [43]. From these data, the proliferation index was calculated as the quotient of the percentage of proliferating cells (sum of events at the S and G2/M gates) and cells in the G0/G1 phase [44].

### 2.8. Assessment of the Endotoxin Content

The chromogenic Limulus Amebocyte Lysate (LAL) kit was used as a quantitative assay for the detection of Gram-negative bacteria endotoxins in the NIST1648a and LAp120 samples, used according to the manufacturer’s instructions. LPS from *E. coli* serotype O111:B4 was used as a reference endotoxin. LPS was dissolved in endotoxin-free water and used at final concentrations of 0.01, 0.1 or 1 μg/mL. Poly B was added to half of the 96-well plates (for a final concentration of 100 μg/mL) to block the biological effects of LPS by binding to lipid A, the toxic component of LPS [45]. The neutralizing effect of Poly B on LPS is dose dependent and specific. Briefly, a limulus polyphemus proenzyme, from the blood of horseshoe crabs, was activated to form a clot after exposure to LPS. Then it catalyzed the splitting of p-nitroaniline from the colorless substrate. After stopping the reaction, the released product was photometrically measured at 405 nm [46]. The developed color intensity was proportional to the amount of endotoxin present in the sample and was calculated using a standard curve.

### 2.9. Isolation of RNA

RNA was isolated using TRIzol reagent (ThermoFisher Scientific, Waltham, MA, USA), following the manufacturer’s protocol, and was further purified using the RNeasy Mini Kit (Qiagen Inc.). The total RNA concentration was measured using a NanoDrop ND-1000 (NanoDrop Technologies Inc., Montchanin, DE, USA). The RNA quality was determined using an Agilent 2100 Bioanalyzer (RIN > 8) (Agilent, Palo Alto, CA, USA).

### 2.10. Microarray Hybridization

The quality of RNA was determined using the RNA 6000 Nano LabChip Kit and Agilent Bioanalyser 2100 (Agilent, Palo Alto, CA, USA). Preparation of complementary RNA (cRNA) was performed according to the protocol provided by Affymetrix (Santa Clara, CA, USA). Total RNA was converted to double-stranded cDNA using the SuperScript System (Invitrogen, Carlsbad, CA, USA) and an oligo(dT)24 primer containing a T7 RNA polymerase promoter site. Biotin-labeled cRNA was synthesized from cDNA using a labeling kit and purified using a GeneChip Cleanup Sample Module (Qiagen Inc., Valencia, CA, USA). The yield of the in vitro transcription reaction was determined by the product absorbance at 260 nm as measured by the NanoDrop ND-1000 (NanoDrop Technologies, Inc., Montchanin, DE). The size of the cRNA probes was evaluated using the RNA 6000 Nano LabChip Kit (Agilent, Palo Alto, CA, USA). Fragmented cRNA was used for hybridization to GeneChip^®^ Mouse Gene Atlas array strips (Affymetrix). The arrays were washed and stained with streptavidin–phycoerythrin (Merck, Darmstadt, Germany) in the GeneAtlas Fluidics Station (Affymetrix), according to the standard protocol provided by the manufacturer. The arrays were scanned using a GeneAtlas Scanner 3000 (Affymetrix). Raw microarray data were generated using the GeneAtlas instrument control system (Affymetrix). A total of 48 Mouse Gene Atlas Affymetrix microarrays were used, with 4 biological replicates per each experimental point.

### 2.11. Microarray Data Analysis

All statistical analyses were performed with the R software v.3.3.1. (available online https://www.r-project.org/, accessed on 1 July 2022). The “Oligo R” package was used for importing CEL files (Affymetrix microarray data) and data preprocessing, which included background subtraction, quantile normalization, summarization and log2-transformation at the gene level. Gene annotations were obtained from the “mogene21stprobeset.db” R package. The signal was taken as the measure of mRNA abundance derived from the level of gene expression. A two-way ANOVA (for the treatment and time factors) was used for the main statistical analysis, followed by the Bonferroni correction for multiple testing. Hierarchical clustering was performed using correlation as the distance measure. The gene annotation tool, Enrichr, was used to identify overrepresented ontological groups among the gene expression patterns and to group the genes into functional classifications [47]. Overrepresented terms (KEGG 2021 Pathways) were defined as having at least two transcripts and a *p-*value < 0.1 per adjusted Fisher’s exact test.

### 2.12. Statistical Analysis

All assays were performed at least in triplicate. Data are presented as the mean ± SEM. Statistica 10.0 for Windows (Statsoft, Tulsa, OK, USA) was used for the analysis of the data. The normality of the variable distributions and the homogeneity of the variances were verified by the Shapiro–Wilk and Levene’s tests, respectively. Data were evaluated by a two-way analysis of variance (ANOVA), with PM and concentration as the factors, separately, for each time point, and by Tukey’s post hoc test, when appropriate. When ANOVA assumptions were not fulfilled, the Kruskal–Wallis ANOVA (by ranks) for multiple comparisons was used. *p*-Values lower than 0.05 were regarded as significant.

## 3. Results

### 3.1. The Effect of PM on Cell Viability and Metabolic Activity

Changes induced by exposure to NIST1648a or LAp120 in RAW 264.7 cell viability in culture, as assessed by supravital PI staining and flow cytometry, are shown in Figure 1. Short-term (6 h) exposure to both forms of PM at a concentration of 100 µg/mL resulted in a slight (up to approximately 90%) but significant (*p* < 0.01) reduction in the percentage of viable cells, with the LAp120 effect being more pronounced (*p* < 0.001 vs. NIST1648a). After long-term exposure, the highest applied concentration of PM caused a significant (up to approximately 70%, *p* < 0.001) decrease in the percentage of viable cells in the culture (Figure 1A), with the effect of NIST1648a being stronger (*p* < 0.001 vs. LAp120). Reduction of cell viability was related to an increase in the percentage of late apoptotic and necrotic cells, with necrosis being more pronounced after short-term exposure (Figure 1B,C). At this time point, the percentage of necrotic events increased parallel to the concentration of PM (*p* < 0.01 and *p* < 0.001 at a higher concentration of NIST1348a and LAp120, respectively; Figure 1C), whereas the percentage of late apoptotic events was significantly increased only by a higher concentration of LAp120 (*p* < 0.001; Figure 1B). The effect of LAp120 on the percentages of both apoptotic and necrotic events was more pronounced than NIST1648a at a concentration of 100 µg/mL (both at *p* < 0.05; Figure 1B,C). After long-term exposure (Figure 1B), apoptosis was significantly increased by both forms of PM, with the effect of NIST1648a being more pronounced than in the case of LAp120 at a higher concentration (*p* < 0.001). Regarding the percentage of dead cells at this time point, the concentration’s main effect was significant only at F(2, 18) = 169.16; *p* < 0.001, where only higher concentrations of both forms of PM increased it at *p* < 0.001 (Figure 1C).

Apoptotic cell death was also assessed cytometrically in fixed cells on the basis of DNA content as the relative number of events in the sub-G1 range. As shown in Figure 2, apoptosis assessed using this method was significantly changed by exposure to both forms of PM (concentration’s main effect: F(2, 12) = 62.66; *p* < 0.001); it decreased with lower concentrations of PM (*p* < 0.01) and increased due to the exposure to higher concentrations of both NIST1648a and LAp120 (*p* < 0.001). Long-term exposure to a higher concentration of NIST1648a increased apoptosis (*p* < 0.05), while exposure to LAp120 did not induce significant changes in comparison to nonexposed cultures (Figure 2).

The metabolic activity, measured based on the resazurin reduction method, was unchanged following short-term (4 h) exposure to both forms of PM (Figure 3). Long-term (48 h) exposure to higher concentrations of NIST1648a or LAp120 resulted in a significant (up to approximately 80 and 60%, *p* < 0.01 and 0.001, respectively) reduction in metabolic activity, with the LAp120 effect being more pronounced (*p* < 0.001 vs. NIST1648a).

### 3.2. The Effect of PM on Cell Proliferation and Cell Cycle Distribution

Disturbances in cell proliferation induced by exposure to NIST1648a or LAp120 are shown in Figure 4. The ratio of the percentage of proliferating cells (in the S and G2/M phases) to cells in the G0/G1 phase was adopted as a measure of proliferation. Both forms of PM inhibited proliferation after 6 h of exposure in a concentration-dependent manner (concentration’s main effect: F(2, 12) = 187.86; *p* < 0.001; with either concentration at *p* < 0.001 as assessed with the post hoc test) and increased it after 48 h (concentration’s main effect: F(2, 18) = 61.10; *p* < 0.001; 1 and 100 µg/mL at *p* < 0.01 and 0.001, respectively); however, after a shorter (6 h) exposure time, LAp120 had a stronger effect (PM’s main effect: F(1, 12) = 5.09; *p* < 0.05; Figure 4).

The above-described changes in cell proliferation were related to the cell cycle disturbances presented in Figure 5. Short-term exposure to NIST1648a or LAp120 increased the percentage of cells in the G0/G1 phase (concentration’s main effect: F(2, 12) = 51.01; *p* < 0.001; with either concentration at *p* < 0.001, as assessed with the post hoc test; Figure 5A) and resulted in a decrease in the relative number of cells in the phase S (concentration main effect: F(2, 12) = 33.05; *p* < 0.001; with either concentration at *p* < 0.001; Figure 5B) and in the G2/M phase (concentration’s main effect: F(2, 12) = 11.49; *p* < 0.01; 1 and 100 µg/mL at *p* < 0.05 and 0.01, respectively; Figure 5C). On the other hand, long-term exposure to both forms of PM caused a concentration-dependent decrease in the proportion of cells in the G0/G1 phase (concentration’s main effect: F(2, 18) = 18.48; *p* < 0.001; only 100 µg/mL at *p* < 0.001; Figure 5A) and an increase in the relative number of cells in the S phase (concentration’s main effect: F(2, 18) = 6.94; *p* < 0.01; only 100 µg/mL at *p* < 0.05; Figure 5B) and in the G2/M phase (concentration’s main effect: F(2, 18) = 78.06; *p* < 0.001; 1 and 100 µg/mL at *p* < 0.01 and 0.001, respectively; Figure 5C). The described changes may indicate a cell cycle arrest in the G0/G1 phase after a short-term (6 h) exposure period and in the G2/M phase after a long-term one.

### 3.3. The Effect of PM on NO Release

Stimulation of RAW 264.7 cells with LPS resulted in a concentration-dependent increase in NO release, assessed as the concentration of nitrites in the culture medium after a 48 h culture period (Appendix A).

The synthesis of NO due to the fact of exposure to NIST1648a or LAp120 was relatively low, and the results are presented in Figure 6 as the percentage of the control values. However, the increase in NO synthesis in cultures exposed to a higher concentration of NIST1648a was statistically significant at the *p* < 0.05 level. Taking into account the reduced viability of cells in culture over 48 h of exposure (up to less than 60%), the actual NO release by the remaining viable cells was even higher. Exposure to LAp120 did not change the concentration of nitrites in the culture medium.

To confirm the above results, the synthesis of NO was investigated using the DAF-2 DA fluorescent probe and microplate reader in control cultures and cultures exposed for 72 h to a higher concentration of NIST1648a or LAp120 (Figure 6B). This assay measures the current production of NO by living, metabolically active cells. This method also showed that only NIST1648a, excluding LAp120, increased NO synthesis. Again, this increase was not large but statistically significant (*p* < 0.05).

### 3.4. The Endotoxin Content in Both PM Forms

As shown in Appendix A, an LPS concentration of 10 ng/mL, which was used as a positive control in the cultures intended for gene expression testing, corresponded to a concentration of just over 2 EU measured with the LAL method. The addition of Poly B at a concentration of 100 µM completely neutralized even the highest concentration of LPS used for cell stimulation in the present study.

The endotoxin content in both forms of PM is presented in Figure 7. At a concentration of 1 µg/mL, it was below the measurement linearity. Activation of the LAL enzymatic system by NIST1648a at a concentration of 100 µg/mL corresponded to approximately 0.9 EU, and the addition of Poly B significantly (*p* < 0.001) lowered the readings to approximately 0.64 EU. Activation of the LAL enzymatic system by LAp120 amounted to approximately 0.4 EU, and the addition of Poly B did not change this value (Figure 7). As the ability to activate LAL is also shown by other organic components potentially contained in PM, such as glucans [48], lipoteichoic acid and peptidoglycan [49], only the part of the LAL activity inhibited by Poly B can be considered as the endotoxin content, since Poly B specifically neutralizes lipopolysaccharides of Gram-negative bacteria [45]. For NIST1648a at a concentration of 100 µg/mL, it would correspond to approximately 0.28 EU/mL, while LAp120 seemed to not contain any LAL-detectable amounts of endotoxin.

### 3.5. Changes in Gene Expression

For the identification of functional links between genes with similar expression profiles, the Enrichr gene list enrichment analysis tool was used. In general, gene expression changes caused by NIST1648a at a higher concentration were similar to those caused by LPS, but. they were slightly less pronounced. The changes induced by LAp120 at a higher concentration were similar in direction but much less expressed and incomparably lower than the effects of NIST1648a. Lower concentrations of both forms of PM had almost no effect on gene expression compared to the control cultures.

This analysis allowed us to distinguish seven gene clusters. Examples of the most overrepresented molecular pathways are presented in Figure 8. A complete list of the terms for each cluster can be found in Appendix A. Cluster 1 included 177 transcripts. Genes in this cluster showed a significant increase in expression after short-term exposure to a higher NIST1648a concentration and LPS, and these changes decreased after long-term exposure. Based on the KEGG pathway database, we assigned regulated genes to 92 functional terms, mainly related to immune response and cell death, all of which were statistically significantly altered (at adjusted *p* < 0.1). Cluster 2 contains seven transcripts that showed a decrease in abundance levels after long-term exposure. Cluster 3 included 65 transcripts. This group, similar to cluster 1, showed an increase in the expression levels of genes as a result of short-term exposure to a higher concentration of NIST1648a and LPS, but the increased expression was still maintained after 48 h of exposure. The genes in this group were associated with 10 terms involved mainly in inflammatory processes and the immune response, all of which were statistically significantly altered (adjusted *p* < 0.1). Cluster 4 contained 14 genes related to one term involved in arginine biosynthesis, enriched at adjusted *p* < 0.1. The genes in this cluster showed increased expression levels in response to a higher concentration of NIST1648a and LPS only after 48 h. Cluster 5 contained three genes activated in response to a higher concentration of NIST1648a and LAp120 at two analyzed time points. The genes in this cluster were associated with ferroptosis, iron metabolism and redox balance. Cluster 6 included 138 transcripts. The genes in this group showed a decreased expression level due to the fact of a higher concentration of NIST1648a and LPS for 4 h of exposure and remained at a reduced level also after 48 h. This group was enriched in genes involved in molecular pathways related to cell cycle regulation, cell senescence and cell death. Cluster 7 contained 108 transcripts. These genes were almost unaffected by a higher concentration of NIST1648a and showed slightly decreased expression in response to LPS after 4 h of exposure, and the decrease in their expression deepened after 48 h. The latter effect was the most intensive under the influence of NIST1648a. Cluster 7 included genes assigned to eight terms related primarily to the cell cycle, DNA replication and proliferation, all of which were statistically significantly altered (adjusted *p* < 0.1).

### 3.6. The Effect of PM or LPS on the Number of Regulated Genes

Similarities and differences in the number of transcripts changed by exposure to both forms of PM and to LPS after 4 and 48 h are shown in Figure 9 in the form of Venn diagrams. For a complete list of genes regulated by exposure to both forms of PM and stimulation by LPS, see Appendix A.

After short-term exposure, the greatest number of genes was regulated in response to LPS treatment—a total of 797. NIST1648a at a higher concentration affected a total of 329 genes, while LAp120 only 19 genes. Among the genes regulated by NIST1648a, 20 were specifically altered only by this form of PM, 307 genes were also regulated by LPS, including 16 also regulated by LAp120. Two genes were altered only by NIST1648a and LAp120. There was no gene exclusively regulated by LAp120 at this time point (Figure 9A).

After long-term exposure, the greatest number of transcripts was changed due to the fact of a higher concentration of NIST1648a—a total of 274. LAp120 modified the expression of 31 genes and LPS that of 244 genes. Among the genes regulated by NIST1648a, 78 were altered only by NIST164a, 189 genes were also regulated by LPS, including 20 also by LAp120 and 7 genes modified only by NIST1648a and LAp120. At this time point, LAp120 specifically regulated only three genes (Figure 9B).

### 3.7. Association of the Changes Observed in Gene Expression and in Assessments of Cell Death

As shown in Figure 1, both forms of PM decreased cell viability and increased mortality, depending on the concentration and exposure time. However, NIST1648a-induced mortality involved the apoptotic pathway to a greater extent than that induced by LAp120, which was also reflected in the degree of DNA fragmentation (Figure 2) and metabolic activity (Figure 3). These data correlate with the expression of genes altered by higher concentrations of NIST1648a or LAp120 contained in clusters 1 and 5 (Figure 8, Appendix A) and assigned to terms associated with different forms of programmed cell death: apoptosis, necroptosis and ferroptosis. For example (Table 1), exposure to both forms of PM mainly triggered the extrinsic apoptotic pathway and necroptosis as indicated by the increased expression of *Tnf* and *Fas*. The expression of these genes was higher under the influence of NIST1648a, both after 4 h (*p* < 1.50 × 10^−5^ and *p* < 9.20 × 10^−5^, respectively) and 48 h of exposure (*p* < 9.60 × 10^−5^ and *p* < 0.0250, respectively) than under the influence of LAp120 (after 4 h: *p* < 0.0016 and *p* < 0.0225; after 48 h: *p* < 0.0409 and *p* < 0.8011, respectively). As a result of exposure, the intrinsic pathway of apoptosis was also activated as indicated by the enrichment of the term p53 signaling pathway (Appendix A).

The contribution of ferroptosis was less diversified depending on the form of PM and may be related to the inorganic component, as the expression of some genes assigned to this term (e.g., Slc40a1 after 4 h of exposure) was higher after LAp120 (*p* < 3.31 × 10^−4^) than after NIST1648a (*p* < 0.0028) exposure, and LPS stimulation had little effect on the gene expression of this term (Table 1, see also Appendix A).

### 3.8. Association of the Changes Observed in Gene Expression and in Cell Cycle Distribution

As shown in Figure 5, exposure to both forms of PM resulted in disturbances in the cell cycle. These disturbances may suggest a cell cycle arrest in the G0/G1 phase after a short-term (6 h) exposure period and in the G2/M phase after a long-term one. They were reflected in the changes induced by a higher concentration of NIST1648a and Lap120 in the expression of the genes contained in clusters 6 and 7 (Figure 8, Appendix A) and assigned to cell-cycle-related terms. For example, Table 2 summarizes some of the most highly altered genes in terms called cell cycle assigned to clusters 6 and 7. A complete list of genes regulated by exposure to both forms of PM or by LPS can be found in Appendix A. Table 2 shows that after short-term exposure, genes involved in both the initial phase, G1, (e.g., *Cdkn2c* from cluster 6) and the later phases of the cell cycle (e.g., *Mcm6* from cluster 6 and *Mad2l1* from cluster 7) were the most highly altered. Long-term exposure resulted in more pronounced changes in genes associated with the later phases of the cell cycle, i.e., S and G2/M (e.g., *Cdc20* and *Ttk* from cluster 6 and *Bub1b* and *Ccnb2* from cluster 7). The changes in gene expression were greater after exposure to NIST1648a than to LAp120, but these differences were not reflected in the percentage of cells in each phase of the cell cycle as shown in Figure 5.

### 3.9. Association of the Changes Observed in Gene Expression and Inflammatory Activation of the Cells

As shown in Figure 6 and referring to NO synthesis, only NIST1648a, excluding LAp120, exhibited proinflammatory activity. This activity seemed to be proportional to the endotoxin content (Figure 7) and was reflected in the changes in the expression of the two main genes involved in arginine metabolism, *Nos2* and *Arg2*, and related to the functional phenotypes of macrophages, M1 and M2 [24] (Table 3). The product of the *Nos2* gene (cluster 1, gene assigned to 11 terms related to infectious and neoplastic diseases), inducible NO synthase (iNOS), is a key enzyme for the synthesis of NO and a marker of the activity of the classical pro-inflammatory subpopulation of M1 macrophages. The product of the *Arg2* gene (cluster 4, arginine biosynthesis), arginase 2, is a key enzyme in the alternative pathway of arginine metabolism and is considered to be a marker of the activity of alternative macrophage (M2) subpopulations with a general anti-inflammatory phenotype [24]. Arginase exists as two isoforms, arginase 1 and arginase 2, which are encoded by different genes, have different tissue and cellular localizations and have different regulatory mechanisms, but they have similar functions. Both forms of arginase are involved in the anti-inflammatory activity of macrophages and their differentiation toward the M2 functional phenotype [50]. In this study, only the expression of *Arg2* was investigated; however, considering the functional similarity of both isoforms, inference regarding its role in shaping the anti-inflammatory phenotype of macrophages seems to be justified. As shown in Table 3, exposure to NIST164a increased *Nos2* expression after 4 h, and *Arg2* activity increased at both time points tested. However, the NIST1648a-induced increase in *Nos2* expression was substantially lower than that induced by LPS. Exposure to LAp120 did not affect the expression of both of these genes (Table 3, Appendix A).

The results shown in Figure 6, illustrating the pro-inflammatory activation of RAW 264.7 cells by NIST1648a as measured by NO synthesis, do not reflect substantial changes in the expression of other genes contained in clusters 1 and 3, and assigned to terms related to pro-inflammatory activity and immune response (Figure 8, see also Appendix A); comparable to the effects of LPS. As an example, Table 3 lists the genes assigned to the most severely altered terms from clusters 1 and 3: the TNF signaling pathway and the IL-17 signaling pathway, respectively.

## 4. Discussion

In this study, the impact of crude PM material, NIST1648a, and that with a reduced organic matter content, LAp120, on selected parameters of cell viability, cell cycle distribution and metabolic and inflammatory activity as well as the gene expression profile in RAW 264.7 macrophages was investigated. Both forms of PM were shown to negatively affect cell viability, metabolic activity, cell proliferation and cell cycle distribution. These results were generally reflected in the gene expression profile. Only NIST1648a, excluding LAp120, contained endotoxins and showed small but statistically significant pro-inflammatory activity. However, gene expression profiling revealed a strong pro-inflammatory cell activation induced by a higher concentration of NIST1648a, especially clearly visible after 4 h of exposure, which was close to the LPS effects. Changes in gene expression due to the fact of a lower concentration of NIST1648a and LAp120 at both concentrations were relatively small.

The enhancement of cell death by various processes, including apoptotic and necrotic ones, represents one of the major negative effects of PM reported in the literature (reviewed by Peixoto et al. [51]). Depending on the cell type, type and concentration of PM and the duration of exposure, one or the other form of cell death is predominant. However, on the basis of the research conducted in this study, it is not possible to precisely indicate the form of cell death, as in some conditions apoptosis leads to secondary necrosis [52].

In the present experiment, concentration-dependent decreases in viability were observed after short-term exposure to both forms of PM, mainly as a result of the necrotic form of cell death with no change or only a slight increase in the percentage of late apoptotic cells in the case of a higher LAp120 concentration as assessed with supravital PI staining [34]. In contrast, the assessment based on DNA content [41] indicated a decrease in apoptosis after short-term exposure to lower concentrations of PM and an increase after exposure to higher concentrations. Differences in the assessment of apoptosis using these methods may result from the fact that the DNA content in living and necrotic cells after fixation is the same [53].

The decrease in cell viability noted in this experiment did not affect the metabolic activity, as the loss of activity caused by the death of some cells can be compensated by increased resazurin metabolism by living cells in response to PM-induced oxidative stress [54]. In our previous studies, an increase in metabolic activity was noted after short-term exposure to a low PM concentration [23]. Similarly, an increase in metabolic activity, as measured by the MTT test, after short-term (2–4 h) exposure was observed by Happo et al. [31]. Other authors also reported a slight increase in resazurin reduction after 48 h of exposure to a low concentration of vanadyl sulfate (VOSO4), which at the same time significantly increased the synthesis of free radicals [55].

As resulting from the research carried out in this study, clearer differences in the effects of NIST1348a and LAp120 were revealed after a longer (48 h) period of exposure. Lower concentrations of PM neither reduced the viability nor changed the metabolic activity. Undoubtedly, these concentrations caused some cell death, as shown in an elegant form by Ardon-Dryer et al. [56], but these losses may be compensated for by enhancing resazurin metabolism in response to PM-induced stress (as mentioned above). Higher concentrations of both forms of PM caused a significant decrease in viability; in the case of NIST16348a, it was largely associated with apoptosis and in the case of LAp120 with necrotic death. These changes were accompanied by a decrease in metabolic activity, less marked in the case of NIST1648a, as apoptotic cells were also metabolically active. Similar results were obtained by others studying the effects of PM and its typical components, carbon black—as a model of pollution related to the anthropogenic sources—and kaolin—as a natural mineral component. In that study, PM decreased the viability of human bronchial epithelial (16HBE) cells mainly by apoptosis, while carbon black and kaolin induced mainly necrotic death [57]. In general, the fate of cells exposed to PM largely depends on the intensity of the oxidative stress induced by exposure, namely, lower ROS synthesis causes apoptotic death and higher ROS synthesis necrotic death (for a review, see [51]). The results obtained in the present study correlate with our previous report [23], which showed that higher cell death after exposure to LAp120 was associated with higher ROS synthesis.

The biochemical manifestations of cell death reported in the present study were generally consistent with changes in gene expression. Exposure to NIST1648a resulted in much more potent activation of genes associated with the two forms of programmed cell death, apoptosis and necroptosis, than exposure to LAp120. The literature indicates that both of these forms occur as a result of exposure to PM [58,59]. Apoptosis can be triggered by external signals as a result of activation of death receptors by appropriate ligands (e.g., FasL/FasR and TNF/TNFR1) or intracellular processes associated with DNA damage or oxidative stress (reviewed in [60]). The results of gene profiling obtained in this study indicated that exposure to both forms of PM induced mainly the extrinsic apoptotic pathway as evidenced by the most pronounced expression of genes of this pathway (i.e., *Tnf*, *Fas* and *Daxx*) and the relatively lower enrichment of the term named the p53 signaling pathway [60,61]. It should be noted, however, that a large part of the reports in the literature indicate the prevalence of the intrinsic apoptotic pathway associated with increased ROS synthesis due to the fact of PM exposure (e.g., [58]).

Necroptosis is a form of cell death that manifests itself similarly to necrosis but goes off in a controlled manner [62]. An increasing number of reports in the literature indicate the participation of this form of cell death in the effects of PM exposure (e.g., [59]). Activation of necroptosis by both forms of PM used in this experiment was similar to apoptosis (i.e., greater changes were caused by exposure to NIST1648a than to LAp120), and *Tnf* and *Fas* were also the most highly altered genes, as both these processes are similar and metabolically related [63,64].

Ferroptosis is yet another form of programmed cell death that manifests itself similarly to necrosis and is induced by oxidative stress associated with disturbances in iron metabolism [65]. As a result of exposure to both forms of PM, the term ferroptosis was significantly enriched, with LAp120 having a slightly greater effect than NIST1648a in contrast to the other forms of cell death discussed above. Moreover, the genes of this term were also assigned to another enriched term called, mineral absorption, and also to terms related to redox balance: glutathione metabolism and the HIF-1a signaling pathway. Furthermore, LPS stimulation had little effect on gene expression attributed to the term ferroptosis. Considering the high iron content in NIST1648a (3.92 ± 0.21%) [32], which is even higher in LAp120 due to the removal of a large proportion of organic compounds as explained below, the obtained results suggest that this form of cell death may be associated with the inorganic component of PM. This suggestion is supported by reports from the literature: both PM [66], iron-bearing nanoparticles [67] and cigarette smoke [68] induce ferroptosis through iron overload, intracellular iron dysregulation and redox imbalance. Similar to our present results, ferroptosis was also among the major pathways in cellular responses to urban PM as reported by Zhu et al. [69].

As in the case of the abovementioned diversity of the pathways of induction of apoptosis by PM, there are discrepancies in the literature regarding the involvement of necrosis and apoptosis in the cytotoxic effect. For example, Ardon-Dryer and colleagues have shown that exposure of the human lung epithelial cell line A549 to low concentrations of montmorillonite particles 2.5 µm in diameter (used as a model of dust storm particles) induced a predominance of apoptotic processes, while higher concentrations led to a predominance of necrotic death [56]. On the other hand, short-term (4 h) exposure of A549 alveolar epithelial cells to biologically active concentrations of cigarette smoke extract (CSE) mainly intensified apoptosis, while 24 h of exposure induced mainly necrotic death [70]. Differences in the results obtained in this experiment and in some reports in the literature may result from differences in the composition of the PM and from different exposure conditions (i.e., exposure time and PM concentration) as well as the cells used and applied methods of mortality assessment. NIST1648a is original atmospheric particulate matter collected in an urban area, with a mean particle diameter of 5.85 µm, in which all components are naturally present [32]. LAp120 is devoid of most organic components as a result of cold plasma treatment, which does not change the physical properties of the PM [23] but increases (by approximately 34%) the percentage of inorganic biologically active components (e.g., metals), because, as a result of removing the organic fraction, the sample mass decreases to approximately 66% [33]. The differences in the intensity of apoptosis and necrosis observed in this study may be related to the content of metals. For example, 24 h of exposure to PM from the combustion of various types of solid fuels increased the apoptotic and necrotic deaths of A549 cells, and the severity of the necrosis was associated with a high metal content [71].

Differences in the chemical composition of the PM also caused variations in gene expression. Examples include studies in which exposure to PM from different sources and with different chemical compositions (e.g., the NIST164b standard or PM collected in Taiyuan in 2017–2018) caused different changes in gene expression [69,72]. Moreover, different extracts of the same PM induced slightly different patterns of gene expression: the aqueous extract induced greater gene changes in molecular pathways related to oxidative stress and inflammation and the organic extract had a greater effect on genes and signaling pathways related to the cell cycle [73].

Numerous data from the literature show that exposure to PM causes disturbances in the cell cycle at various stages. For example, it was shown that 4 h of exposure of alveolar epithelial (A549) cells to PM induced a cell cycle arrest in the G1 phase and inhibited cell proliferation by a decrease in cyclin E, A, D1 and cyclin E-cyclin-dependent kinase (CDK)-2 activity [74]. In another study, 48 h of exposure of two human lung carcinoma cell lines, H292 and H1299, to PM resulted in an S-phase block of the cycle as a result of cyclin D1 downregulation [75]. In turn, other authors have shown that exposure of human bronchial epithelial (BEAS-2B) cells to PM causes DNA damage and cell cycle arrest in the G2/M phase after 24 h as a result of the activation of the Ras/Raf/MAPK pathway and overexpression of c-Myc [76]. The results obtained in this study indicate that short-term exposure to both forms of PM inhibits proliferation through an arrest in the G0/G1 phase, while long-term exposure increases proliferative activity but, at the same time, disrupts the cell cycle by inducing a block in the G2/M phase. These effects were more pronounced in the case of higher concentrations of both PM forms and are generally consistent with the literature cited above.

However, the result obtained by us indicating an increase in proliferation after longer (48 h) exposure does not necessarily reflect an actual increase in cell numbers. In this study, proliferative activity was estimated on the basis of the assessment of DNA content by flow cytometry [41] and expressed as the sum of the percentage of cells with more than 2N DNA content [44]. Such an estimate seems to be justified, as cells with replicating DNA can be considered proliferating. However, this method does not prejudge whether cell divisions were completed correctly, as in some situations cells complete mitosis without division, and such 4N cells are interpreted by a flow cytometer as G2/M cells [77]. Such an effect of PM was described by Longhin et al. [78]. In their experiment, short-term exposure of BEAS-2B cells to PM resulted in a cell cycle arrest in the G2/M phase, which caused a delay in mitosis after 24 h and an increase in the number of tetraploid G1 cells. Other studies have shown that PM reduces cell proliferation, growth [76] and the number of cell divisions as well as extends the time of division [56]. Our recently published results also showed a substantial reduction in cell numbers after long-term exposure to both forms of PM, NIST1648a and LAp120 [23]. Thus, the increase in the percentage of proliferating cells in combination with a cell cycle arrest in the G2/M phase observed in this study would, in fact, be a manifestation of a decrease in cell proliferation. This suggestion is supported by the analysis of changes in gene expression presented in this paper, which distinguished two clusters of genes with reduced expression under the influence of exposure, especially the longer one (48 h). The most highly changed terms in these clusters were related to the cell cycle and DNA synthesis, namely, cell cycle, p53 signaling pathway, oocyte meiosis, DNA replication and cellular senescence. In reports by other authors, as a result of 12 h of exposure of human monocytic THP-1-derived macrophages to PM collected in Beijing, China, the same terms were also among the most highly altered [73].

The experiments conducted in this study showed no differences in the effects of NIST1648a and LAp120 on cell cycle progression and cell proliferation. Some reports in the literature correlated selected PM components with specific effects on the cell cycle. For example, Yang et al. [79] showed correlations between some PAHs, elemental carbon, As, Ni and percentages of cells in the G0/G1 and G1/G2 phases, respectively. On the other hand, other authors presented results suggesting that changes caused by PM in the cell cycle progression may be related to ROS synthesis rather than to specific components [71,78]. As both of the forms of PM used in this study induced strong, although differentiated, synthesis of ROS [23], taking into account the complexity of the processes leading to cell cycle disturbances, the lack of significant differences in the effects of NIST1648a and LAp120 on cell cycle progression is not surprising.

Pro-inflammatory cell activation leading potentially to serious systemic disturbances is one of the major negative effects of PM exposure [19,80,81]. The research carried out in this study showed that only NIST1648a, excluding LAP120, induced small but statistically significant pro-inflammatory activity, manifested by increased NO synthesis. The increase in NO synthesis as a measure of pro-inflammatory activity has been reported in the literature (e.g., [16,82,83,84]). PM-induced NO synthesis was substantially diversified, and in some reports the exposure of cells to PM only slightly increased the accumulation of nitrites in the culture medium (e.g., [84]), while in others, the effect of PM was much stronger, similar to the effect of LPS stimulation (e.g., [82]).

Endotoxins are one of the most biologically important groups of PM components. They are responsible, to a significant extent, for the pro-inflammatory activity of PM, as their neutralization by Poly B significantly reduces the synthesis of pro-inflammatory cytokine, IL-6 and, to a slightly lesser extent, also TNF-α [16]. It has also been shown that the pro-inflammatory activity of PM is positively correlated with the content of endotoxins, both in in vitro [10] and in vivo studies [85]. The endotoxin content in PM varies depending on the emission source and the season, and it is usually related to the coarse fraction, i.e., with a particle size of 2.5 µm or greater [11,14,16]. It ranges from 0.27 to 0.56 EU/mg [10] through 2.43 to 17.8 EU/mg [11] and approximately 10-40 EU/mg [14] up to 138.4-170.4 EU/mg [17]. The endotoxin content in NIST1648a determined in this study was close to the lower value given by Liu et al. [11] and appeared to be proportional to the induced pro-inflammatory activity expressed as NO synthesis. In contrast, LAp120 did not contain any LAL-detectable amounts of endotoxins and did not induce NO synthesis. The above results suggest that bacterial endotoxins may be one of the groups of components responsible for the pro-inflammatory activity of NIST1648a. This suggestion is supported by the results of gene profiling presented in this paper. The gene expression changes induced by NIST1648a were very similar to those induced by LPS, and the most highly altered terms were related to inflammatory processes and immune response. Moreover, only NIST1648a caused changes in the expression of *Nos2* and *Arg2*, two genes related to functional macrophage polarization [24]; however, the increase in *Nos2* expression induced by NIST1648a was substantially lower than that induced by LPS.

The pro-inflammatory activity of NIST1648a measured in this study by the intensity of NO synthesis did not reflect robust changes in the expression of genes related to inflammatory processes and immune response. One of the reasons for this disproportion may be the negative influence of PM on the expression of iNOS [29]. Further explanation may be found in the present experiment, where the pro-inflammatory activity was assessed only on the basis of NO synthesis. Data in the literature indicate that the production of inflammatory cytokines could be a better indicator. For example, Salonen et al. [84] showed that a relatively small increase in NO synthesis as a result of exposure of RAW 264.7 cells to PM (approximately 1.8-fold) was accompanied by a much larger increase in the secretion of TNF-α (over a dozen times) and IL-6 (several dozen times). Thus, although there was a statistically significant positive correlation between NO synthesis and the production of both IL-6 and TNF-α [83], NO synthesis alone may not be an adequate measure of PM-induced pro-inflammatory cell activation. In addition, considering the chemical complexity of PM and the multitude of metabolic pathways involved in the cell response, the disproportion in gene expression and biochemical results does not seem surprising.

Nevertheless, data in the literature indicate that exposure of cells to PM changes the functional phenotype of macrophages towards the pro-inflammatory M1 by increasing *iNOS* expression and reducing *Arg-1* expression, accompanied by an increase in TNF-α release and a decrease in the production of the anti-inflammatory cytokine IL-10 [86]. In other studies, such polarization was manifested by an increase in iNOS, IL-6 and TNF-α expression, both at the mRNA and protein level, and the activation of the TLR4/NF-κB signal transduction pathway, with no changes in the expression of M2 subpopulation markers: ARG1 and CD206 [30].

The pro-inflammatory activity of PM connected with the presence of endotoxins described above is an important mechanism of its negative influence on the body. For example, in vitro studies have shown that exposure of lymphocytes to PM induces the pro-inflammatory activation of these cells, and the endotoxins contained in PM are an important stimulatory agents, as their inactivation by Poly B significantly reduces the pro-inflammatory activity of PM [87]. Numerous data from the literature indicate that exposure to PM induces local inflammatory processes in the respiratory system that lead to systemic changes, and thus can affect distant organs [19]. It is noteworthy to mention its negative effect on the central nervous system and neurodegenerative processes (reviewed by Jankowska-Kieltyka et al. [88]). As a result of PM exposure, in addition to the pro-inflammatory changes in the respiratory system mentioned above, disturbances in the composition of the intestinal flora [89] and an increase in gut permeability occur [90]. This leads to the induction of inflammatory processes in the digestive system and an increase in the penetration of LPS of intestinal bacteria into the circulation and further inflammatory signaling into the brain, which contributes to the pathomechanisms of many neurodegenerative diseases such as Parkinson’s disease, Alzheimer’s disease or multiple sclerosis [91]. A similar mechanism has been described for major depression [92] and other neuropsychiatric disorders [93]. Thus, the LPS/endotoxins, though external factors and not produced by the human body, become signaling molecules that are importantly involved at the nexus of neuropsychiatric and neurodegenerative disorders. Taken together, PM not only contains bacterial endotoxins but also increases their release from the gastrointestinal tract and penetration into the central nervous system, which contributes to disturbances in its functioning and neurodegenerative processes.

In addition to endotoxins, other organic components of PM that are removed by NIST1648a treatment with cold oxygen plasma [33] may also be associated with pro-inflammatory cell activation, consequently causing relatively small changes in the gene expression reported in this article, for example, PAHs, which are ligands of aryl hydrocarbon receptors (AHRs) that exhibit pro-inflammatory effects (for a review, see [94]). It has been shown that PAHs intensify inflammatory processes, but individual components of this group show different effectiveness [95]. Moreover, various components contained in PM exhibit interactions in biological effects (synergism or antagonism), and the effects produced by a single component differ from those exerted by their mixtures [96] as exemplified by the synergistic effect of diesel exhaust particles and minerals on the pro-inflammatory activity measured by gene expression and the synthesis of the following cytokines: IL-8, IL-1β and IL-1α [97]. Thus, apart from endotoxins, other organic and mineral constituents may also be involved in the pro-inflammatory activity of PM observed in the present study.

The research carried out in this paper indicates some molecular mechanisms of the negative impacts of PM on cells, in particular their pro-inflammatory activation. Bearing in mind that macrophages are an important link in the body’s first line of defense against harmful factors and their participation in maintaining homeostasis of various tissues, as well as their potent ability to regulate the immune response [20,21,22], the use of the macrophagal RAW 264.7 cell line seems justified. This cell line is widely used in in vitro research in this field (e.g., [11,16,29,30,31]; see also [28]). Studies in in vitro cell models are relatively quick and simple, and when performed on one well-known cell type, they provide detailed information on the mechanisms involved. However, the use of only one type of cell excludes the influence and interactions of other cells that comprise tissues and organs, and these interactions are extremely important for the proper functioning of organs and the body as a whole. This is particularly important in the case of freshly harvested cells, when only one type of cells is isolated and tested from the whole organism [28].

## 5. Conclusions

The results presented in this study indicate that both NIST1648a and LAp120 exert a negative influence on cells. However, the effects of either form of PM were varied, mainly in the area of pro-inflammatory activity. Particularly marked differences were found at the level of gene expression. The observed differences were related to the content of organic matter in which bacterial endotoxins played an important role. However, other organic compounds (e.g., PAH) as well as their interactions with different components of PM also appeared to be of importance. The pro-inflammatory activity of PM, especially that connected to the bacterial lipopolysaccharides contained within them, is an important mechanism of their negative impacts on the body including the central nervous system and neurodegenerative processes.

## Figures and Tables

**Figure 1 biomolecules-12-01100-f001:**
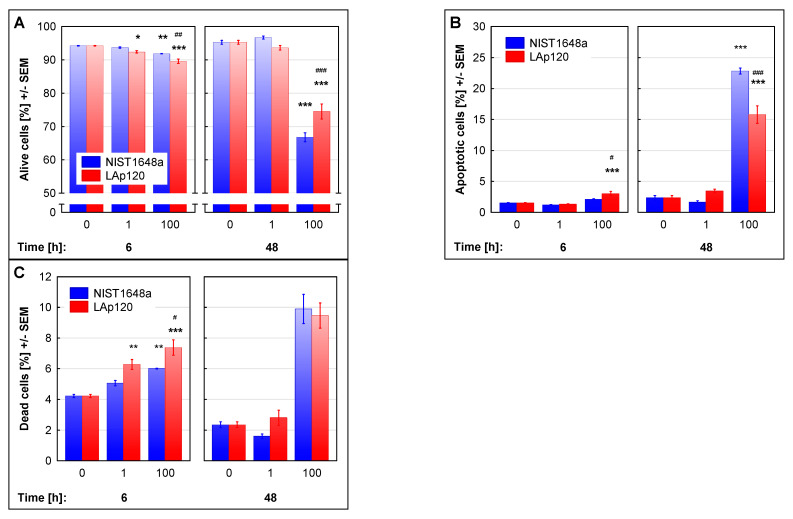
Viability of the cells assessed with supravital propidium iodide (PI) staining and flow cytometry and expressed as the % of live cells: not stained with PI (**A**); dimly stained late apoptotic cells (**B**); brightly stained necrotic cells (**C**) among all events. RAW 264.7 cell cultures were exposed to NIST1648a or LAp120 at concentrations of 0, 1 or 100 µg/mL of the culture medium (abscissa) for 6 and 48 h. * *p* < 0.05, ** *p* < 0.01, and *** *p* < 0.001; relative to the control cultures exposed to neither NIST1648a nor LAp120 (PM = 0 µg/mL) at the same time point. # *p* < 0.05, ## *p* < 0.01, and ### *p* < 0.001; for cultures exposed to NIST1648a vs. LAp120 at the same concentration and time point. After long-term exposure to (**C**) only, the concentration’s main effect was significant at F(2, 18) = 169.16; *p* < 0.001, where the influence of higher concentrations of both forms of PM was significant at *p* < 0.001.

**Figure 2 biomolecules-12-01100-f002:**
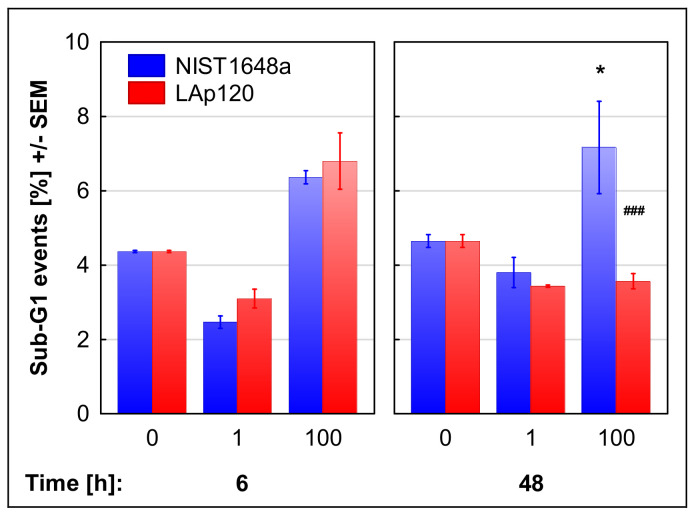
Apoptosis assessed in fixed cells, on the basis of DNA content and measured with a flow cytometer, as the relative number of events in the sub-G1 range. RAW 264.7 cell cultures were exposed to NIST1648a or LAp120 at concentrations of 0, 1 or 100 µg/mL of the culture medium (abscissa) for 6 and 48 h. After 6 h of exposure, the concentration’s main effect was significant only at F(2, 12) = 62.66; *p* < 0.001. A lower concentration (regardless of the PM form) reduced the percentage of apoptotic cells at *p* < 0.01, whereas a higher concentration increased it at *p* < 0.001. * *p* < 0.05; relative to the control cultures exposed to neither NIST1648a nor LAp120 (PM = 0 µg/mL) at the same time point. ### *p* < 0.001; for cultures exposed to NIST1648a vs. LAp120 at the same concentration and time point.

**Figure 3 biomolecules-12-01100-f003:**
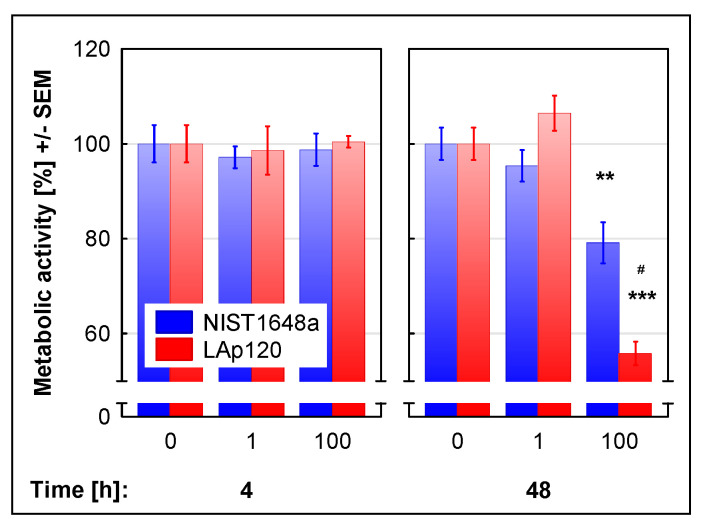
Metabolic activity assessed with the resazurin reduction test and expressed as a % of the control values. RAW 264.7 cell cultures were exposed to NIST1648a or LAp120 at concentrations of 0, 1 or 100 µg/mL of the culture medium (abscissa) for 6 and 48 h. ** *p* < 0.01, and *** *p* < 0.001; relative to the control cultures exposed to neither NIST1648a nor LAp120 (PM = 0 µg/mL) at the same time point. # *p* < 0.05; for cultures exposed to NIST1648a vs. LAp120 at the same concentration and time point.

**Figure 4 biomolecules-12-01100-f004:**
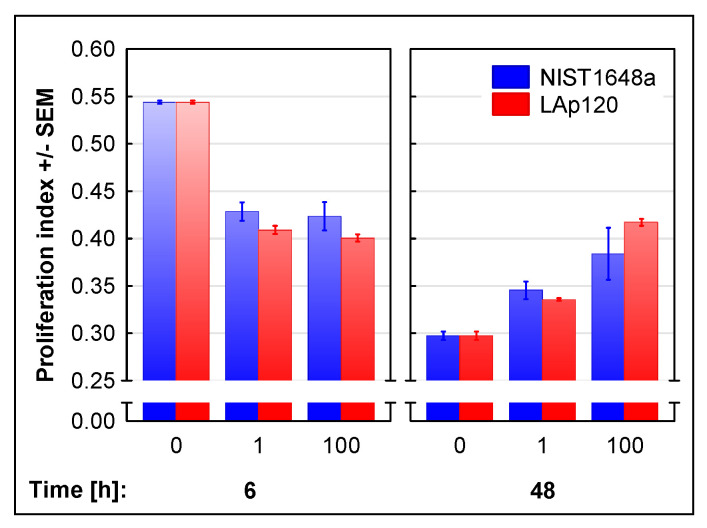
Proliferation of the cells expressed as the quotient of the percentage of proliferating cells (sum of events in the S and G2/M gates) and cells in the G0/G1 phase assessed in fixed cells on the basis of DNA content and measured with a flow cytometer. RAW 264.7 cell cultures were exposed to NIST1648a or LAp120 at concentrations of 0, 1 or 100 µg/mL of the culture medium (abscissa) for 6 and 48 h. After short-term exposure, concentration’s main effect was significant at F(2, 12) = 187.86; *p* < 0.001, with either concentration at *p* < 0.001. In addition, PM’s main significant effect was at F(1, 12) = 5.09; *p* < 0.05. After long-term exposure only, concentration’s main effect was significant at F(2, 18) = 61.10; *p* < 0.001, where the influence of 1 and 100 µg/mL was significant at *p* < 0.01 and 0.001, respectively, regardless of the PM form.

**Figure 5 biomolecules-12-01100-f005:**
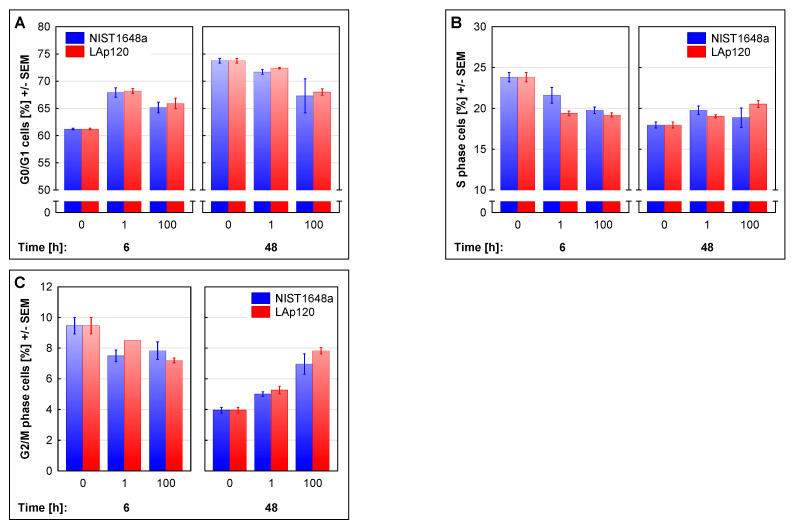
Cell cycle distribution assessed in fixed cells, on the basis of DNA content and measured with a flow cytometer, as the percentage of cells in the G0/G1 (**A**), S (**B**) and G2/M (**C**) phases. RAW 264.7 cell cultures were exposed to NIST1648a or LAp120 at concentrations of 0, 1 or 100 µg/mL of the culture medium (abscissa) for 6 and 48 h. For the G0/G1 phase at the 6 and 48 h time points, concentration’s main effects were significant at: F(2, 12) = 51.01; *p* < 0.001; with either concentration at *p* < 0.001 as assessed with the post hoc test; F(2, 18) = 18.48; *p* < 0.001; only 100 µg/mL at *p* < 0.001, respectively. For the S phase at the 6 and 48 h time points, concentration’s main effects were significant at: F(2, 12) = 33.05; *p* < 0.001; with either concentration at *p* < 0.001; F(2, 18) = 6.94; *p* < 0.01; only 100 µg/mL at *p* < 0.05, respectively. For the G2/M phase at the 6 and 48 h time points, concentration’s main effects were significant at: F(2, 12) = 11.49; *p* < 0.01; 1 µg/mL at *p* < 0.05 and 100 µg/mL at *p* < 0.01; F(2, 18) = 78.06; *p* < 0.001; 1 µg/mL at *p* < 0.01 and 100 µg/mL at *p* < 0.001, respectively.

**Figure 6 biomolecules-12-01100-f006:**
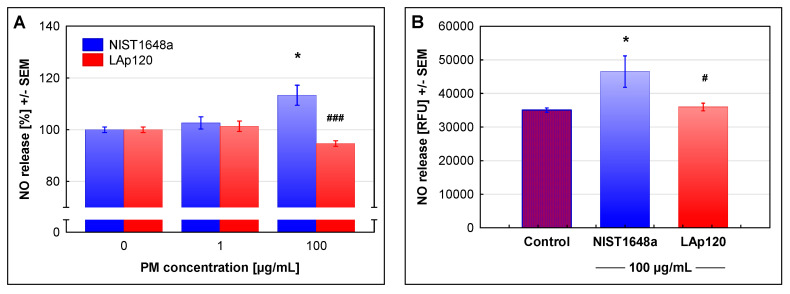
NO release assessed using the Griess reaction and expressed as the percentage of the values observed in the control cultures not exposed to PM. RAW 264.7 cell cultures were exposed to NIST1648a or LAp120 at concentrations of 0, 1 or 100 µg/mL of the culture medium (abscissa) for 48 h (**A**). NO release was assessed, using the 4,5-diaminofluorescein diacetate (DAF-2 DA) fluorescent probe and microplate reader, by the number of viable cells after only 90 min of incubation period and expressed in RFUs. RAW 264.7 cell cultures were exposed to NIST1648a or LAp120 at a concentration of 100 µg/mL of the culture medium (abscissa) for 72 h (**B**). * *p*< 0.05; relative to the control cultures exposed to neither NIST1648a nor LAp120 (PM = 0 µg/mL) at the same time point. # *p*< 0.05, and ### *p*< 0.001; for cultures exposed to NIST1648a vs. LAp120 at the same concentration and time point.

**Figure 7 biomolecules-12-01100-f007:**
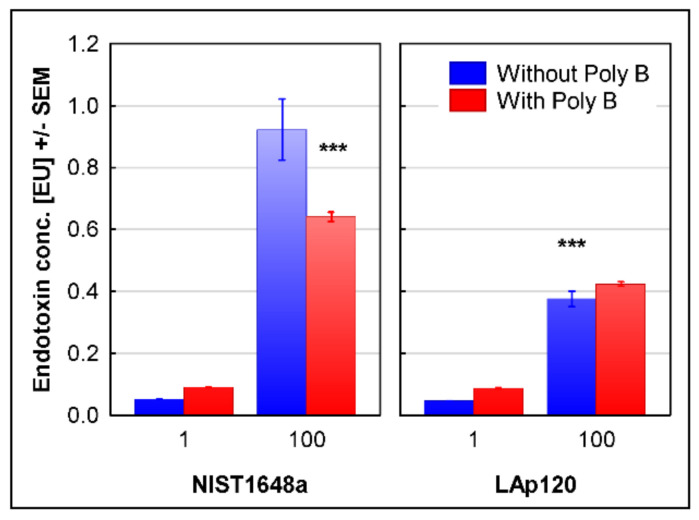
The endotoxin content in NIST1648a and LAp120 at concentrations of 1 or 100 µg/mL of the culture medium (abscissa) as assessed with the LAL method. *** *p* < 0.001 vs. NIST1648a at 100 µg/mL without Polymyxin B (Poly B).

**Figure 8 biomolecules-12-01100-f008:**
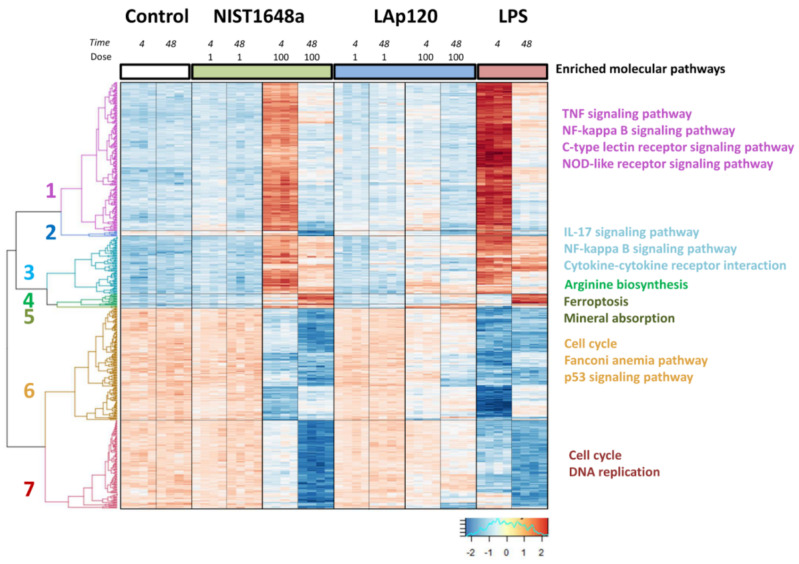
Gene expression patterns regulated in response to PMs in RAW 264.7 macrophages. Microarray results are shown as a heat map and include 535 transcripts with a genome-wide significance from a two-way ANOVA for the treatment factor (corrected *p* < 10^−10^) and the difference between the control vs. LAp or NIST (a fold-change of log2 > 1.5). The colored rectangles represent transcript abundance at 4 or 48 h after treatment with LAp (1 or 100 µg/mL), LPS (10 ng/mL), NIST (1 or 100 µg/mL) or untreated control cells. The intensity of the color is proportional to the standardized values (between −2 and 2) from each microarray, as displayed on the bar below the heat map image. Hierarchical clustering was performed using the Euclidean distance. Major clusters of treatment-responsive genes are arbitrarily described as 1–7. Biological pathways (based on the KEGG 2021 Human dataset) enriched in genes (adjusted *p* < 0.05) from a particular cluster (indicated by the cluster color) are presented on the right.

**Figure 9 biomolecules-12-01100-f009:**
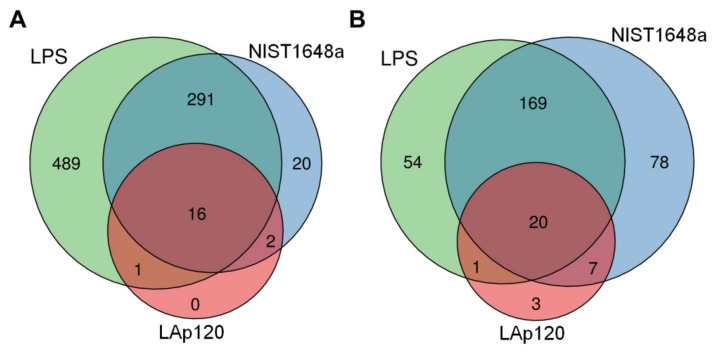
Comparison of the number of genes with the expression altered by the PMs and LPS in RAW 264.7 macrophages 4 (**A**) and 48 h (**B**) after treatment. Sixteen transcripts were shared between genes regulated 4 h after the treatment with NIST1648a, LPS and Lap120a. Approximately three-quarters of the genes altered by LPS were also regulated by NIST1648a treatment. Genes were identified as regulated using the treatment factor (ANOVA, with a threshold at *p* < 10^−10^) and interaction between treatment and time factors (ANOVA, with a threshold at *p* < 10^−10^). The list of transcripts with a fold-change greater than 1.5 and a *t*-test threshold (*p* < 0.05) between the treatment and control was analyzed.

**Table 1 biomolecules-12-01100-t001:** Changes in gene expression as a result of exposure to NIST1648a or Lap120 at a concentration of 100 µg/mL, associated with different forms of programmed cell death (i.e., apoptosis, necroptosis and ferroptosis) and expressed as the significance of the differences from the controls at respective time points (*p*-values with Bonferroni correction).

Cluster and Term	Gene Name	NIST1648a		LAp120	
		4 h	48 h	4 h	48 h
Cluster 1: Apoptosis	*Birc3*	0.0161	1	0.3763	1
	*Daxx*	1.50 × 10^−5^	1	1	1
	*Fas*	9.20 × 10^−5^	0.0250	0.0225	0.8011
	*Gadd45b*	0.0162	1	1	1
	*Nfkb1*	0.0023	1	0.2698	1
	*Nfkbia*	4.78 × 10^−4^	1	0.0220	1
	*Tnf*	1.50 × 10^−5^	9.20 × 10^−5^	0.0016	0.0409
	*Traf1*	0.0010	1	0.1156	1
Cluster 1: Necroptosis	*Birc3*	0.0161	1	0.3763	1
	*Fas*	9.20 × 10^−5^	0.0250	0.0225	0.8011
	*Jak2*	0.0110	1	1	1
	*Nlrp3*	0.0072	1	0.2144	1
	*Stat2*	0.0251	1	1	1
	*Tnf*	1.50 × 10^−5^	9.60 × 10^−5^	0.0016	0.0409
	*Tnfaip3*	1.10 × 10^−4^	1	1	1
	*Zbp1*	5.46 × 10^−4^	1	1	1
Cluster 5: Ferroptosis	*Gclm*	0.0016	0.0041	0.0135	0.0204
	*Hmox1*	0.0266	0.0139	0.0462	0.0169
	*Slc40a1*	0.0028	0.3026	3.31 × 10^−4^	0.0159

**Table 2 biomolecules-12-01100-t002:** NIST1648a- or LAp120-induced changes in the expression of genes assigned to the most severely altered terms (i.e., cell cycle) included in clusters 6 and 7 related to the cell cycle and expressed as the significance of the differences from the controls at respective time points (*p*-value with Bonferroni correction). Cell cultures were exposed to both forms of PM at a concentration of 100 µg/mL for 4 or 48 h.

Cluster and Term	Gene Name	NIST1648a		LAp120	
		4 h	48 h	4 h	48 h
Cluster 6: Cell cycle	*Bub1*	0.1272	6.71 × 10^−4^	1	1
	*Ccna2*	0.0826	0.0023	0.1973	1
	*Ccne2*	0.0036	4.30 × 10^−5^	0.2351	1
	*Cdc20*	1	5.70 × 10^−5^	1	0.0011
	*Cdc6*	0.0948	6.20 × 10^−4^	0.979	1
	*Cdkn2c*	7.72 × 10^−4^	0.0032	0.0536	0.7181
	*Chek2*	0.0033	0.3525	1	1
	*Mcm6*	2.70 × 10^−5^	1.8. × 10^−4^	0.2187	1
	*Mcm7*	0.0309	0.0042	1	1
	*Plk1*	0.0691	6.00 × 10^−5^	1	0.0711
	*Tfdp2*	9.21 × 10^−4^	0.0062	0.2147	1
	*Ttk*	0.0044	3.90 × 10^−5^	1	0.0485
	*Wee1*	0.0178	3.58 × 10^−4^	0.4424	1
Cluster 7: Cell cycle	*Bub1b*	0.0247	4.50 × 10^−5^	1	0.0054
	*Ccnb1*	0.1038	0.0311	1	1
	*Ccnb2*	1	0.0027	1	0.0352
	*Cdc25c*	0.0513	0.0150	1	1
	*Cdk1*	1	0.0118	1	1
	*Espl1*	1	0.0365	1	1
	*Mad2l1*	0.0087	0.0040	0.1029	1
	*Mcm5*	0.1021	0.0074	1	1

**Table 3 biomolecules-12-01100-t003:** NIST1648a- or LAp120-induced changes in the expression of genes assigned to the most severely altered terms from clusters 1 and 3, associated with inflammatory and immune activity as well as arginine metabolism pathways, and expressed as the significance of the differences from controls at respective time points (*p*-value with Bonferroni correction). Cell cultures were exposed to both forms of PM at a concentration of 100 µg/mL for 4 or 48 h.

Cluster and Term	Gene Name	NIST1648a		LAp120	
		4 h	48 h	4 h	48 h
Cluster 1	*Nos2*	0.0471	0.1388	1	1
Cluster 3	*Arg2*	0.0492	0.0151	1	1
Cluster 1: TNF signaling pathway	*Birc3*	0.0161	1	0.3763	1
	*Ccl5*	9.73 × 10^−4^	0.0128	1	1
	*Csf1*	1.10 × 10^−5^	0.0152	0.0086	1
	*Csf2*	3.80 × 10^−5^	1	1	1
	*Cxcl2*	7.00 × 10^−6^	0.0248	9.20 × 10^−4^	1
	*Cxcl3*	3.98 × 10^−4^	0.2441	1	0.2036
	*Edn1*	0.0410	1	1	1
	*Fas*	9.20 × 10^−5^	0.0250	0.0225	0.8011
	*Icam1*	2.93 × 10^−4^	1	0.2066	1
	*Il6*	9.69 × 10^−4^	0.0942	1	1
	*Irf1*	0.0555	1	1	1
	*Junb*	0.1259	0.0666	1	1
	*Lif*	0.0020	1	1	1
	*Lta*	0.0021	0.6914	0.9670	1
	*Nfkb1*	0.0023	1	0.2698	1
	*Nfkbia*	4.78 × 10^−4^	1	0.02120	1
	*Ptgs2*	4.90 × 10^−5^	3.09 × 10^−4^	0.0014	1
	*Tnf*	1.50 × 10^−5^	9.60 × 10^−5^	0.0016	0.0409
	*Tnfaip3*	1.10 × 10^−4^	1	1	1
	*Traf1*	0.0010	1	0.1156	1
Cluster 3: IL-17 signaling pathway	*Ccl2*	3.59 × 10^−4^	0.0034	1	1
	*Ccl7*	0.0149	6.07 × 10^−4^	1	1
	*Lcn2*	2.64 × 10^−4^	0.0547	0.6715	1
	*Mmp9*	0.0202	1	0.6668	1

## Data Availability

Data are contained within the article.

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
