# Peer review of "Gene Expression Changes Induced by Exposure of RAW 264.7 Macrophages to Particulate Matter of Air Pollution: The Role of Endotoxins"

_biomolecules, 2022, doi:10.3390/biom12081100_

Round 1

Reviewer 1 Report

This is a report about gene expression changes induced by exposure of RAW 264.7 macrophages to particulate matter and identify the role of endotoxins by a series of methods. There were some very interesting and promising results. However, the report should improve the English generally. Also, the authors had done detailed discussion comparing the PM samples and LPS, especially the differences in the biological endpoints on inflammation and oxidative stress. There should be some discussion regarding pros and con of the cellular models herein mouse macrophages used for pulmonary toxicity assessment and the results in comparison to other similar studies in the literature. The paper will also benefit taking the following remarks into account.

1.       Please generate a list of Abbreviations. Also, PM is short for particulate matter. Please adjust this in the abstract and use it properly through the whole manuscript.

2.       Please improve the significance labelling and description for all figures, and add the significance for those missing labelling. For example, for Figure 1, description should cover *, ##, ### as well. For legend for figure 2, please use the same format in terms of description of the * and ###.

3.       It is not necessary to have supplementary figures in the main texts. Please remove them.

4.       Please clarify Figure S2.

5.       It would be interesting to know the further studies following this study.

Reviewer 2 Report

Line 18: Space between 264.7 and cells.

Line 64: Why are neutrophils excluded here as they are innate immune cells, express TLR4, and are abundant in lungs during pollution exposure.

Line 96: Proper nomenclature dictates E. coli in italics (throughout the manuscript).

Lines 89 & 122-123: Are these doses and times representative of environmental exposures of these compounds? Or were levels and times used that only induced significant expression changes? This is a question of relevance.

Line 145: How were the macrophages removed for flow?

Line 214: Delete space between 96 -well.

Figure 1: The statistical significance indicators in the figure legend are confusing. What does * indicate? What does ### indicate? Also, in Fig 1C, no stat sig in the 48 hr time point for 100 vs the other concentrations? Same issue for other figures in the manuscript. 

The selection of 0, 1, and 100 ug/mL seems odd to find effects of dosing. Was a 10 ug/mL dose not assayed?

Why was LPS not used as a control in the cell death and metabolic activity assays?

Figures 4 and 5: Why are no stat sig indicators used on the figures?

Why are Supplemental Figures included in the main text? These are by definition then not supplemental. Also, what is the difference between “Supplemental Figure S2” and “Figure S2” on page 11?

Figure 7 is discussed in the text but not labeled in any figure. It is likely Figure S2.

No stats consideration for correlations in Supplemental Figure 1? Supp Figure 2?

How is exposure of PM in Figure 6B a higher concentration as described. Is it not the same 100 ug/mL? The time changed to 72 hrs.

Figure S2 (or Figure 7) information would be valuable direct contribution to the cell survival, proliferation, and metabolic assays performed earlier, for example, Poly B in addition to NIST1648a as a control.

Lines 425-426: Would this not be obvious even before the experiments?

The authors should justify the microarray data collected at 4 hr and not 6 hr as all of the other experiments. Comparing Figure 8 gene expression results at different time points (4 hr vs 6 hr) in earlier figures does not seem appropriate unless clearly justified. Also, Figure 8 legend states 4 and 28 hr, not 48 hr as shown in the graphic.

Proper nomenclature dictates that all genes be italicized (throughout manuscript)

Lines 549-564: As written, it appears that the authors are suggesting that Arg2 plays a role in M1-like macrophage function when it is Arg1 known to drive this activity. Arg2 has actually been suggested to be biased toward an M1-like macrophage function. Could be typo error, but all references throughout the manuscript should be revised. Also in Line 801.

Line 657: Missing a period between “[59] As a result…”

Line 661: Should be “HIF-1a

Lines 657 and 663: “Ferroptosis” should not be capitalized.

The discussion is a little lengthy.

Any connection to macrophage functions from the microarray outside of the 48 hr time point for matched stimulus is only speculative.

RAW264.7 cells are mouse macrophages originally obtained from ascites fluid (abdominal source). The high plasticity of macrophages due to their tissue environments and stimuli suggests that the authors should justify their selection of these cells for pulmonary macrophage experiments and production of factors reported detrimental to health. Primary mouse lung macrophages could be obtained by bronchoalveolar lavage to provide more impact to this study.

Reviewer 3 Report

I don't this manuscript (even though interesting and well written) fits well to this special issue entitled: "Signaling Biomolecules in the Central Nervous System (CNS)—At the Nexus of Neuropsychiatric and Neurodegenerative Disorders", there is only one paragraph that relates the studies topic to nervous system. I suggest to reconsider this manuscript for a regular submission or a different special issue more focused on the effects of particulate matter on respiratory system as it is written now. 

Here are few other comments that should be addressed before the manuscript is accepted for publication: 

-          It would be helpful if authors express the exposure doses used per surface area of exposed cells, and extrapolate this dose to the surface area of human lungs.

- Figure 6. Add plots in which NO release is normalized for viable cells only

- Iron content in both samples should also be expressed as mass concentration per per weigh, for easier comparison with other studies and between samples.

- Supplementary Figure S2. - numbers should be corrected using periods not commas

Round 2

Reviewer 2 Report

I still have some reservations about macrophage sources/cell lines being utilized to represent and extrapolate functions in actual tissues, especially in tissues of different origin. But the authors do cite sufficient evidence that these cells are used commonly in the PM field. I congratulate the authors for their meticulous responsiveness to my rather rigorous review. The paper is much better.